# Deep Brain Stimulation in the Subthalamic Nucleus Can Improve Skilled Forelimb Movements and Retune Dynamics of Striatal Networks in a Rat Stroke Model

**DOI:** 10.3390/ijms232415862

**Published:** 2022-12-14

**Authors:** Stefanie D. Krämer, Michael K. Schuhmann, Jens Volkmann, Felix Fluri

**Affiliations:** 1Radiopharmaceutical Sciences/Biopharmacy, Institute of Pharmaceutical Sciences, Department of Chemistry and Applied Biosciences, ETH Zurich, 8093 Zurich, Switzerland; 2Department of Neurology, University Hospital Würzburg, Josef-Schneider Strasse 11, 97080 Würzburg, Germany

**Keywords:** photothrombosis, experimental stroke, subthalamic nucleus, invasive electric stimulation, skilled forelimb movements, neuronal network, [^18^F]FDG positron emission tomography

## Abstract

Recovery of upper limb (UL) impairment after stroke is limited in stroke survivors. Since stroke can be considered as a network disorder, neuromodulation may be an approach to improve UL motor dysfunction. Here, we evaluated the effect of high-frequency stimulation (HFS) of the subthalamic nucleus (STN) in rats on forelimb grasping using the single-pellet reaching (SPR) test after stroke and determined costimulated brain regions during STN-HFS using 2-[^18^F]Fluoro-2-deoxyglucose-([^18^F]FDG)-positron emission tomography (PET). After a 4-week training of SPR, photothrombotic stroke was induced in the sensorimotor cortex of the dominant hemisphere. Thereafter, an electrode was implanted in the STN ipsilateral to the infarction, followed by a continuous STN-HFS or sham stimulation for 7 days. On postinterventional day 2 and 7, an SPR test was performed during STN-HFS. Success rate of grasping was compared between these two time points. [^18^F]FDG-PET was conducted on day 2 and 3 after stroke, without and with STN-HFS, respectively. STN-HFS resulted in a significant improvement of SPR compared to sham stimulation. During STN-HFS, a significantly higher [^18^F]FDG-uptake was observed in the corticosubthalamic/pallidosubthalamic circuit, particularly ipsilateral to the stimulated side. Additionally, STN-HFS led to an increased glucose metabolism within the brainstem. These data demonstrate that STN-HFS supports rehabilitation of skilled forelimb movements, probably by retuning dysfunctional motor centers within the cerebral network.

## 1. Introduction

Approximately 70–80% of all stroke survivors have upper limb (UL) motor impairments, and 60% of them show incomplete functional UL recovery 6 months after stroke [1,2]. A decrease in strength, loss of motor control and impaired dexterity are frequent and involve difficulties in grasp, object manipulation and tool use [3,4]. These impairments of upper limb following stroke are classically attributed to focal tissue damage [5,6,7]. However, the brain is considered as a complex system of networks consisting of functionally distinct regions that provide both local processing within and distributed processing across regions [8,9,10,11]. Hence, focal stroke lesions involving either cortical and subcortical regions or descending fiber tracts may impair neural activity in this network and disturb even brain areas remote from the ischemic lesion [12,13,14]. This disruption in structural and functional connectivity has been shown to play a key role in poststroke deficits and recovery [10,14,15]. In order to retune the affected interplay between cortical and subcortical regions after stroke, neuromodulation by deep brain stimulation (DBS) might be a promising approach to improve UL motor deficits. A well-established target structure for DBS is the subthalamic nucleus (STN) [16,17,18]. DBS of the STN has been shown to be highly effective in treating dysfunctional motor network activity in Parkinson’s disease [19,20] and seems to be promising in dystonia [21,22]. Additionally, the STN has been proposed as a stimulation target in epilepsy [23,24], non-parkinsonian tremor, such as essential tremor [25], as well as in obsessive-compulsive disorders [26] and substance abuse disorders [27]. The STN is a lens-shaped diencephalic nucleus composed of mainly glutamatergic neurons [28] that serves as an interface between hierarchically higher- and lower-ordered cerebral regions [28,29]. In primates, the STN has been divided into three anatomic–functional regions, i.e., sensorimotor, limbic and associative region [30,31]. The STN receives its main afferents from the basal ganglia (external globus pallidus), motor cortex (hyperdirect pathway) and substantia nigra [32,33,34,35]. On the other hand, most efferent projections of the STN end in the internal globus pallidus [36,37]. Functionally, the STN is a key modulator of basal ganglia output activity [38,39,40] and is therefore important for motor control. High frequent deep brain stimulation (HFS) of the STN is already an established method to improve motor symptoms of Parkinson’s disease [41].

Whether STN-HFS also improves motor deficits of upper limb following stroke has not been addressed so far. Furthermore, studies on costimulated brain regions during STN-HFS are sparse [42]. In this context, 2-[^18^F]fluoro-2-deoxyglucose ([^18^F]FDG) positron emission tomography (PET) is an imaging method allowing quantification of regional cerebral glucose metabolism with respect to neural activity and thus enables the investigation of neuronal networks as well as functional connectivity in the brain [43,44]. Additionally, several studies report on regional changes of cerebral metabolism when DBS is applied [45,46,47,48]. Thus, showing differences in cerebral glucose metabolism before and after STN-HFS might allow costimulated brain regions and networks to be determined.

With these considerations in mind, (i) we investigated whether continuous STN-HFS for 7 days improves forelimb use after photothrombotic stroke in male Wistar rats using a single-pellet reaching task and (ii) determined costimulated brain regions during STN-HFS by calculating differences between [^18^F]FDG-uptake of distinct cerebral areas before and after applying STN-HFS.

## 2. Results

### 2.1. Determination of Electrode Placement and Lesion Volume

During surgery, a microelectrode recording-guided implantation was used to target the tip of the electrode in the STN. When advancing the electrode dorsoventrally in the brain, the firing pattern of STN neurons was identified in all animals between 7.7 to 8.1 mm measured from the surface of the dura (Figure 1A). Correct placement of the electrode in the STN was seen on T2w scans in vivo (Figure 1B) and was furthermore verified on HE-stained brain slices in all animals (Figure 1C). Most tips of electrodes were localized within the STN.

Two rats of group 1 showed macroscopically no photothrombotic lesion when the brain was removed from the skull and therefore were excluded from further analyses. All other animals among group 1 and 2 revealed a photothrombotic stroke encompassing the forelimb motor cortex contralateral to the preferred paw (Figure 1D). In group 1, volume of photothrombotic stroke was quantified using T2w imaging on day 7 after intervention (Figure 1E). Comparison of volume sizes did not yield a significant difference between stimulated and sham-stimulated animals (lesion volume sham stim vs. stim (mean ± SD): 76.4 ± 34.2 mm^3^ vs. 54.9 ± 21.2 mm^3^) (Figure 1F).

### 2.2. Single-Pellet Reaching Test (SPRT)

Single-pellet reaching is a well-established task for examining fine motor control in which rats have to reach, grasp and retrieve a food pellet through a small shift before eating it [49]. Two rats were excluded from analyses because they did not successfully grasp and retrieve pellets within 10 min in the last SPRT before intervention.

One day before intervention, successful reaching of pellets within 10 min was similar in both groups dedicated to continuous STN stimulation (*n* = 8) and sham stimulation (*n* = 8) (number of reached pellets: 21.3 ±7.2 vs. 24.4 ±12.8, *p* > 0.05). Among the stimulated group, the determined stimulating amplitude was (mean ± SD) 40.6 ± 20.6 µA; the continuous STN-HFS was performed for 7 days. On day 2 after intervention, 3/8 of the stimulated animals and 2/8 of the sham-stimulated rats showed a successful reaching of pellets, and thus, success rate did not differ significantly between these two groups. However, when comparing success rate of post-interventional day 2 and day 7, a significantly higher success rate was found in stimulated than in sham-stimulated rats (29% vs. 8.6%; *p* = 0.03) (Figure 2).

### 2.3. Cerebral Network Activation after STN-HFS Using PET Analyses

Tracing studies in rats have revealed efferent STN connections to the globus pallidus (external part), entopeduncular nucleus (EP), substantia nigra (SN) [37] and the mesopontine tegmentum (i.e., pedunculopontine tegmenteal neucleus (PTg)) [50]. Furthermore, there is some evidence that the STN provides direct efferents to the sensorimotor and prefrontal cortices in rats [51] and to the spinal cord [52]. On the other hand, the STN receives afferent projections from the motor cortex [53,54], thalamus [55] and ventromedial nucleus of the hypothalamus (VMH) [56] as well as from brainstem nuclei [57,58]. Based on these anatomical STN connections, glucose metabolism was assessed in the following cerebral circuits and brain regions of rats: (i) pallido-subthalamic and corticosubthalamic circuit; (ii) association cortices; (iii) brain areas contributing to locomotor behavior such as the PTg and VMH as well as the lateral hypothalamus (LH); (iv) brainstem motor areas. 

Out of six examined animals, four were finally included in the PET analysis. Figure 3 provides an overview of elevated [^18^F]FDG uptake in different brain regions after applying STN-HFS in a rat with photothrombotic stroke. Within the pallidosubthalamic circuit, STN-HFS resulted in a significant increase of glucose metabolism in the ipsilateral striatum, globus pallidus and in the thalamus but not in the substantia nigra. On the contralateral hemisphere, however, no change in glucose metabolism was seen in these brain areas. (Figure 4A,B). The same was also true for the corticosubthalamic circuit: When STN-HFS was applied for 1 h, a significant increase of [^18^F]FDG uptake was observed in the motor and somatosensory cortex ipsilateral to the stimulating site but not in the contralateral hemisphere (Figure 4A,B).

Association cortices that were examined in the present study were the cingulate cortex (Cg), the parietal association cortex (PtA), the medial prefrontal (MPFr) and orbitofrontal (OFr) as well as the retrosplenial cortex (RSD). Additionally, the visual cortex (Vis) and hippocampus (Hip) were also investigated. When animals were stimulated in the STN, almost all association cortices as well as Hip and RSD of the ipsilateral hemisphere showed a significantly higher glucose metabolism. On the contralateral hemisphere, Cg and PtA exhibited a significantly increased [^18^F]FDG uptake, whereas OFr showed a significantly reduced glucose metabolism after stimulation (Figure 4C,D).

After applying STN-HFS, the ipsilateral as well as the contralateral Med showed a significant increase of glucose metabolism. The PTg, however, which is connected to the STN, did not demonstrate a change in [^18^F]FDG uptake when STN-HFS was applied (Figure 4E,F).

[^18^F]FDG uptake was further calculated in the brainstem since electrical pulses activate myelinated fibers also antidromically. Compared to the unstimulated state, STN-HFS resulted in a significant increase of [^18^F]FDG uptake in the brainstem with a predominance of the ipsilateral side (Figure 4G,H). When matching the region of significant increase of [^18^F]FDG uptake in the brainstem with the rat brain atlas, regions of elevated glucose metabolism correlated with the localization of pontin nuclei (PnO, PMnR, MnR) on the ipsilateral side, but also with the ventral part of the medullary reticular nucleus (MdV), spinal trigeminal nucleus (Sp5), parvicellular reticular nucleus (PCRt), spinal vestibular nucleus (SpVe) and vestibular nucleus (Ve), as well as with the gigantocellular nucleus (Gi) (Figure 4G,H).

## 3. Discussion

Impaired hand function is recognized as one of the most frequent long-term deficits after ischemic stroke [59]. In healthy subjects, skilled hand movements are related to coordinated neural networks, which are disturbed after stroke [14]. A promising tool for retuning brain networks is electrical stimulation of distinct brain areas which has been shown to enhance recovery in animal stroke models [60,61,62]. Here we sought to determine the impact of STN-HFS on skilled forelimb reaching and its modulating effect on brain networks in rats undergoing photothrombotic stroke. Our results demonstrated that STN-HFS ipsilateral to the photochemically lesioned sensorimotor cortex improves forelimb dexterity in rats when applied continuously for 7 days and increases significantly glucose metabolism within the corticosubthalamic and pallidosubthalamic pathway as well as in the brainstem as shown by [^18^F]FDG PET imaging. Furthermore, infarct size of stimulated rats was slightly but not significantly smaller compared to that of sham-stimulated animals in the present work. Of note, some studies have shown that recovery of motor symptoms after experimental stroke depends on lesion size and is—to some extend—mediated by the spared neurons in the periinfarct region of the motor cortex representing the forelimbs [63,64,65]. In addition, the ability to recover from lesion-induced impairments is linked to the location of lesion; damage of both the caudal and rostral forelimb brain area impairs grasping, whereas a lesion within the caudal forelimb area leads to an impairment of spontaneous forelimb use [65]. Thus, one might argue that the slight difference of infarct size might bias motor outcome between these two groups. However, an intra-group comparison of stimulated animals did not yield a more favorable success rate in animals with smaller infarcts; the same was true for sham-stimulated rats. Furthermore, photothrombotic stroke in all animals encompassed both, the caudal and rostral forelimb area.

Interestingly, when switching on STN-HFS, skilled forelimb grasping did not improve immediately in rats after cortical stroke. Improved forelimb movements were only seen after a seven-day period of continuous STN-HFS. Similar to our observation, STN-HFS in hemiparkinsonian rats resulted in a progressive recovery of skilled forelimb reaching only after a 3- or 5-week period of electrical stimulation [66,67,68]. These findings indicate that recovery of motor symptoms after STN-HFS may be due to neuroplasticity in the perilesional motor cortex. Spieles-Engemann and coworkers [69] have demonstrated that 6-hydroxydopamine (6-OHDA) lesioned rats show a significant improvement of motor forelimb function when applying STN-HFS for two weeks. In addition, they found an increase of brain-derived neurotrophic factor (BDNF)-protein within the nigrostriatal system as well as bilaterally in the motor cortex (M1). These data are underpinned by a recently published study that revealed a significant increase of BDNF mRNA in the ipsilateral striatum and contralateral cortex of 6-OHDA lesioned rats after STN-HFS for 14 days [70]. The upregulation of BDNF-protein after continuous STN-HFS suggests that this treatment may promote a neuroplastic process particularly in the perilesional motor cortex M1 and thus results finally in motor recovery. This effect of STN-HFS is probably mediated by stimulating signals travelling either along the direct subthalamo-cortical projections, namely to the orofacial motor cortex M1 [51], or antidromically along the cortico-subthalamic interconnections. The latter was confirmed by several studies revealing that STN-HFS in parkinsonian rats results in antidromically activated cortical layer V and/or layer II/III neurons in the motor cortex (M1), which was detected by recording of cortical evoking potentials or local field potentials [71,72,73,74]. Consistent with these findings of cortical activation, our analyses of [^18^F]FDG PET scans revealed an increase in glucose metabolism in the perilesional sensorimotor cortex which reflects an activation of these structure due to STN-HFS and thus might corroborate the hypothesis of neuroplasticity mediated by STN-HFS.

Many previous studies investigating behavior in animal stroke models have focused on the role of cortical structures (in particular motor cortex M1) as primary substrates for recovery of skilled forelimb movements [75,76,77,78]. However, there is growing evidence, that execution of dexterous forelimb function and thus, recovery from stroke relies on a coordinated interplay between the primary motor cortex M1 and a distributed cortical and subcortical motor network encompassing the thalamus, basal ganglia, brainstem and spinal cord [79,80,81]. Here, by using an approach of metabolic connectivity, we show that STN-HFS during stimulation results in an increase of glucose metabolism not only in the perilesional motor cortex, but also in the somatosensory cortex, thalamus and striatum as well as in the brainstem. This observation demonstrates that cortical and subcortical areas remained probably functionally and structurally connected after photothrombotic stroke. Furthermore, these findings are similar to those of a [^18^F]FDG-PET study on unilateral STN-HFS in the naïve (un-lesioned) rat brain using a constant current mode at 130 Hz and an amplitude of 300 µA [42]. That study revealed also a significantly increased [^18^F]FDG uptake in the brainstem, cingulate cortex ipsi- and contralateral as well as in the mediodorsal thalamus, the globus pallidus and the caudate nucleus ipsilateral to the stimulation site. However, in contrast to our study, the aforementioned work [42] found a reduced [^18^F]FDG uptake in the somatosensory cortex ipsi- and contralateral as well as in the hippocampus ipsilateral to the stimulation site. These differences may result from the different amplitudes used in these two studies (300 µA vs. 40.6 µA). Additionally, they used naïve rats, whereas our study investigated STN-HFS in animals with photothrombotic stroke. Additionally, neuroimaging studies have provided changes in functional connectivity across cortical motor areas [82,83] and among cortical and subcortical brain regions [84] immediately after stroke that are not found in healthy animals.

When HFS was delivered into the STN ipsilateral to the photothrombotic stroke, a significant increase of [^18^F]FDG uptake was seen in the ipsilateral striatum. In the intact brain, the dorsolateral striatum is thought to modulate vigor and speed of movements [85,86] and—along with corticostriatal interactions—is pivotal for learning and control of skilled forelimb movements [87,88]. After stroke, the neural activity of the dorsolateral striatum is affected in a manner that rats still perform reaching attempts (i.e. touching the food pellet) but are no longer capable to grasp [79]. Interestingly, changes of neural activity in perilesional motor M1 cortex and dorsolateral striatum occurred simultaneously in rats and became more precisely correlated during rehabilitation, which in turn correlated with motor recovery [79,88]. In the present study, the increase of glucose metabolism in both the striatum and the perilesional cortex as well as the improvement of skilled forelimb use after STN-HFS (without further rehabilitation) may point to a STN-HFS-related precise temporal coordination of neuronal activity patterns between these two brain structures.

In the present study, [^18^F]FDG PET analyses yielded a higher glucose metabolism in the rostral and caudal medulla of rats after cortical stroke when STN-HFS was applied. In rats, the STN receives projections from the PTg, the reticular nuclei and the dorsal raphe nucleus [57,58,89] along which DBS stimuli propagate to the brainstem. Within the motor network, the brainstem represents a junction for processing signals from upper motor centers involved in planning actions and circuits in the spinal cord which enables executions of locomotion and forelimb movements [90,91]. The rostral medulla—in particular the lateral part—is believed to regulate forelimb use [80,92]. The rostral medulla encompasses neuronal ensembles projecting mainly to forelimb-innervating but not hindlimb-innervating motor neurons [92]. Interestingly, different sets of neurons within the rostral medulla have different axonal targets and are recruited in a task-specific manner during forelimb movements when elicited by optogenetic stimulation [80]. Neuronal ensembles projecting to the spinal cord are involved in reaching movements, whereas neuronal populations with projections to the caudal medulla are active during more complex forelimb movements such as digit-engaging food-handling movements or grooming [80]. In the present study, increased glucose metabolism was also detected in the caudal medulla which corresponds anatomically with the MdV as well as with the SP5 and thus with two nuclei exhibiting a strong connectivity to spinal forelimb motor neurons [92]. Excitatory neurons in the MdV are an important neuronal substrate for processing information to generate complex forelimb movements, especially for the grasping phase [92]. Moreover, the gigantocellular nucleus (Gi), which is linked to both, fore- and hindlimb, showed also a significant glucose uptake during STN-HFS, which may contribute to an improved forelimb grasping.

There is some evidence that distinct areas of the hypothalamus may impact motor behavior. A previous study has demonstrated that hypothalamic orexin/hypocretin neurons localized in the lateral hypothalamus are involved in skilled movements and especially in error-based motor adaption [93]: After compromised or even lost motor task, the so-called “internal model”, understood as the entirety of cortical motor commands, uses the discrepancy between the desired and present movement trajectory to currently update motor command across attempts and thus reduces error [93]. In the present study, STN-HFS did not change glucose metabolism in the LH, however, resulted in a decrease of [^18^F]FDG uptake in the VMH.

We are aware of the following limitations: First, the number of rats is small and therefore the present study is by its nature explorative. Second, the brain regions were drawn from an atlas and not individually localized in the brains. Third, several of the analyzed brain regions are of small size and may be below the spatial resolution of preclinical PET. The Inveon μ-PET scanner used in this study has a resolution below 2 mm. The signals may furthermore be affected by partial volume effects and spill over from adjacent regions. These results, in particular the reported changes of glucose uptake in the small brain nuclei, should therefore be interpreted with caution. Fourth, a single behavioral test with respect to forepaw motor function might not completely prove the effect of STN-HFS on recovery of forepaw impairment. Thus, other tests such as the ladder rung walking test, pasta handling task or cylinder test should be included in further studies. Nevertheless, the SPRT is a widely used and well-evaluated test, which yielded robust results in this exploratory study.

In conclusion, STN-HFS ipsilateral to the dominant, photochemically lesioned brain hemisphere might promote neuroplasticity by triggering BDNF-expression in the motor cortex and in turn provides a learning-based reacquisition of compromised skilled use of forelimbs. Additionally, STN-HFS may support impaired motor adaption, which finally results in an improvement of pellet grasping and retrieving. Thus, STN-HFS might be an interesting therapeutic approach for treating patients with UL paresis after a stroke. Based on the behavioral test used in this study, our [^18^F]FDG-PET findings indicate that STN-HFS might retune neuronal networks involved in UL motor function and corroborate recent studies that impairments after ischemic stroke are due to distributed brain circuit disruption.

## 4. Materials and Methods

### 4.1. Animals

Adult 8- to 10-week-old male Wistar rats (*n* = 26; Charles River, Sulzfeld, Germany), weighing 321 ± 14 g at the time of intervention, were used in this study. Rats were housed in standardized cages in groups of three in a room maintained on a 12h light (7:00 a.m.–7:00 p.m.)/12 h dark cycle with controlled temperature and humidity. Before starting experiments, rats were acclimatized for 7 days and allowed free access to food and water for this period.

All animal experiments were approved by the institutional review board of the Julius-Maximilians-University, Würzburg, as well as the local authorities of the Regierung von Unterfranken, Würzburg, Germany (TVA55.2-2531.01-102/13) and were conducted in accordance with the ARRIVE guidelines and the EU Directive 2010/63/EU.

### 4.2. Experiment Design

In the present study, rats were assigned to two different groups. Group 1 (*n* = 20 animals) was trained in the single-pellet reaching test (SPRT) for 4 weeks. Then, rats underwent photothrombotic stroke of the motor cortex and electrode implantation in the STN. Thereafter, they were equally divided into a stimulated and sham-stimulated group. Continuous STN-HFS and sham stimulation was started 24 h after intervention and was conducted for 7 days. During this period, SPRT was carried out in stimulated and unstimulated rats on day 2 and day 7 after intervention. After completing the last experiment, cranial magnetic resonance imaging (MRI) was performed in each rat in order to determine infarct volume. Then, rats were euthanized under deep anesthesia and brains were harvested and frozen immediately.

Group 2 (*n* = 6) was only subjected to photothrombotic stroke of the motor cortex and electrode implantation in the ipsilateral STN. On day 2 after intervention, [^18^F]FDG-PET imaging was carried out without and on day 3 with STN-HFS (Figure 5). Dividing animals into these two groups was necessary since each of the two PET measurements required an anesthesia of 60 min. When animals were awake thereafter, a complex behavior test was no longer possible because of an anesthesia-related drowsiness over hours.

### 4.3. Training and Testing of Skilled Forelimb Movements

Deficits and functional recovery of skilled forelimb use can be quantified in a reach-to-eat paradigm such as the SPRT. The test apparatus (modified from Whishaw and Pellis [49]) consists of a transparent Plexiglas box (L36 × W18 × H35.5 cm) with a vertical slit (1 cm wide, 31.5 cm long, beginning 4 cm from the floor) in the center of each front wall (custom-made, GT-Labortechnik, Arnstein, Germany). On the outside of each wall, a shelf (W18 cm) with two indentations was attached in front of the slit (4 cm above the floor). The indentations, each holding one food pellet (45 mg dustless precision pellet, product-# F0021, Bio-Serv, Frenchtown, NJ, USA), were located 1.5 cm from the inside of the wall aligned with the edges of the slit. This prevents a rat from reaching the pellet with its tongue.

Training of the single-pellet reaching started 4 weeks before intervention; the training and test sessions took place daily between 9:00 a.m. and 1:00 p.m. for 10 min. Initially, food pellets were put into both the right and left indentation of the shelf fixed on the front walls. Each rat was trained (i) to reach for a pellet through the slide of the front wall, to retrieve and to bring it to its mouth and eat it; (ii) to leave the slide after eating the pellet, walk to the rear wall, grasp and eat the pellet and finally turn and approach again the slide of the front wall for the next pellet. Once a rat has demonstrated a preference for one paw, by making more attempts to grasp with it, pellets were only put into the indentation contralateral to that paw—a procedure which enforces its use.

The reaching success of each rat was scored for a maximum of 10 min. during which the rat could grasp for at least 20 pellets [94]. A score of 1 was given if the rat was able to reach the pellet, retrieve and bring it to the mouth and eat it. If the rat dropped the pellet inside the test box before bringing it to its mouth or if the rat knocked the pellet off the shelf without grasping and retrieving it, 0 points were awarded. The success rate of each rat was calculated by summing up its pellet scores on postinterventional day 2 and day 7, respectively, which then were divided by the last scores before intervention multiplied by 100. To evaluate the effect of STN-HFS on impaired skilled reaching after photothrombotic stroke, the success rate of day 7 was normalized to that of day 2 after intervention.

In order to increase the motivation of rats to grasp and eat the pellets, they underwent a food restriction beginning one day before the first training and, thereafter, for the 4-week training period. For this reason, rats were daily weighed and chow amount was calculated accordingly so as to maintain body weight at 90–95% of the level usually expected in their age (in compliance with the growth chart of Wistar rats).

### 4.4. Photothrombotic Stroke

Induction of photothrombotic stroke was carried out in rats as described elsewhere [60]. In brief, under deep anesthesia (isoflurane 2.5%), the head of the rat was fixed in a stereotactic frame. After a midline incision through the scalp, the skin laps were displaced laterally in order to expose the surface of the skull. Then, a stencil of aluminum foil leaving out a rectangular aperture (10 × 5 mm) was put on the skull above the primary motor cortex including the forelimb area (5 mm anterior to 5 mm posterior and 0.5 mm to 5.5 mm lateral, relative to Bregma) [95]. The photothrombotic stroke was inflicted contralateral to the dominant/preferred forelimb as determined by success scores during the SPR training or on the right side in rats intended to PET scans. Among the stimulated group, 4 rats preferred the right and 4 the left paw, whereas in the sham-stimulated group, 3 animals showed a preference for the left and 5 for the right paw. A fiber-optic bundle (9 mm diameter) connected to a cold light source (Olympus KL1500LCD, Mainz, Germany) was positioned over the aperture. A total of 0.5 mL Rose Bengal (Sigma-Aldrich, St. Louis, MO, USA) in NaCl 0.9% (10 mg/mL) was injected intravenously within 2 min., before the circumscribed cerebral area was illuminated through the intact bone for 15 min. During the surgery, body temperature was maintained at 37 ± 0.5 °C by a feedback-controlled heating system.

### 4.5. Microelectrode Implantation

Immediately after induction of photothrombotic stroke, a monopolar microelectrode made of platinum/iridium (catalogue# UE-PSEGSECN1M; FHC Inc., Bowdoin, ME, USA) with a mean impedance of 1.09 ± 0.09 MΩ was implanted into the STN contralateral to the dominant paw as described elsewhere [96]. Briefly, a hole for the stimulating microelectrode was drilled 3.6 mm posterior and 2.5 mm lateral relative to Bregma, corresponding with the anterior–posterior localization of the STN [95]. After being attached to a micromanipulator (ROE-200, Sutter Instrument Company, Novato, CA, USA), the microelectrode was inserted into the borehole and lowered to the STN, which is located 7.5 to 8.1 mm dorsoventral from dura and relative to Bregma [52]. At a depth of 7 mm, extracellular recording of single unit action potentials of STN neurons was performed by means of Patchmaster software (HEKA Elektronik GmbH, Lambrecht, Germany v2.65) and a signal amplifier (ELC-03XS, NPI Electronic GmbH, Tamm, Germany) (modified from Maesawa et al. [97]). Firing of STN neurons can be clearly identified since the STN demonstrated high-frequency, irregular firing patterns, whereas surrounding structures such as the zona incerta or cerebral peduncle are relatively silent [97]. Thus, this procedure allows the tip of the stimulating electrode to be placed more precisely in the STN. Once the tip of electrode was located in the STN, the electrode was attached on anchor screws fixed in the skull by using dental cement. A plug (GT-Labortechnik, Arnstein, Germany) was connected with the electrode and fixed with additional dental cement. Wound edges were sutured, and then the animals were allowed to wake up.

### 4.6. Stimulation of Subthalamic Nucleus

In the present study, high-frequency stimulation (HFS; 130 Hz; pulse length, 60 µs; monophasic square wave pulses) was applied into the STN using a stimulus generator (Multichannel Systems, Reutlingen, Germany). Threshold current amplitude was assessed individually for each animal by beginning with 20 µA and then increasing the intensity in steps of 10 µA until dyskinesia were recognized. Then, the electric intensity was reduced by 10–20% below the intensity that elicited dyskinesia or until neurologic signs disappeared and the animal was comfortable. The intensity determined in this way was used for continuous STN-HFS as well as for STN-HFS during [^18^F]FDG-PET imaging.

Continuous HFS of the STN (STN-HFS) started on day 1 after intervention and was applied 24 h a day for 1 week. For this purpose, each rat was housed in a single cage. An electrical cable was plugged into the head plug of the rat, and the other end was connected with a stimulus generator (STG-4004; Multichannel Systems, Reutlingen, Germany). In contrast to the stimulated group, the stimulus generator remained off in sham-stimulated rats (i.e., control group).

### 4.7. Radiochemistry

[^18^F]Fluoride for the synthesis of [^18^F]FDG was produced on the GE-PETtrace cyclotron (General Electric Medical Systems, Uppsala, Sweden) at the Interdisciplinary PET-Centre (IPZ) of the University Hospital of Würzburg via ^18^O(p, n)^18^F reaction by irradiating 3.0 mL of 97% enriched [^18^O]H_2_O with 16.5 MeV protons. [^18^F]FDG was synthesized by a standard procedure, using a GE Fastlab^®^ synthesis module equipped with a FDG-Fastlab^®^ cassette (GE Medical Systems, Uppsala, Sweden) as previously described [98]. [^18^F]FDG was dissolved in injectable NaCl 0.9% for the experiment.

### 4.8. Small Animal Positron Emission Tomography (PET)

PET imaging was carried out on day 2 after intervention without STN-HFS and on day 3 with applying HFS into the STN. Before each PET scan, rats underwent a food deprivation for about 12 h, with free access to water.

μ-PET scans were performed in animals using the Inveon μ-PET scanner (Inveon^®^, Siemens Medical Solutions, Knoxville, TN, USA) as published recently [44]. Rats were anesthetized with isoflurane (1.5%) and body temperature was maintained at 37 °C with a custom-made heating pad during the whole experiment. In the scan session 1 (photothrombotic stroke, no STN-HFS) and scan session 2 (photothrombotic stroke, STN-HFS), rats received an injection of 46.5 ± 17.3 MBq and 48.0 ± 12.6 MBq of [^18^F]FDG, respectively, in a volume of 450–600 µL into the tail vein. Immediately after injection of [^18^F]FDG, the animal was placed in a prone position on the bed of the PET scanner. PET acquisition was started between 3 and 6 min after tracer injection for a duration of 60 min, followed by a transmission scan. During the emission scan (list-mode), STN-HFS was applied for 60 min according to the study protocol. An OSEM 3D algorithm was employed for reconstruction using the PET scanner default settings and including attenuation correction from the transmission scans but without scatter correction.

### 4.9. Evaluation of Glucose Metabolism in Different Brain Regions

Reconstructed PET data were analyzed with PMOD v3.6 software (PMOD Technologies Ltd., Zurich, Switzerland) as described elsewhere [44] and with R (R-project; v4.1.3) using the package oro.nifti. Data were averaged for the complete scan duration and matched to a brain atlas [95] from which TIFF images were imported into PMOD. Volumes of interest were drawn manually based on the imported atlas. PET kBq/cm^3^ were normalized to the averaged kBq/cm^3^ of the cerebellum. Cerebellum was chosen for normalization as its standardized uptake value (SUV), as kBq/cm^3^ normalized to injected dose in kBq per g body weight showed the lowest variability comparing all scans. Changes in [^18^F]FDG uptake are expressed as ((ratio of cerebellum-normalized activity between stimulated and non-stimulated) − 1) × 100%. The matching of the PET images to the brain atlas and subsequent data analyses were performed several times, and the matching was controlled by a second investigator to exclude errors from imprecise matching.

### 4.10. Measurement of Photothrombotic Lesion Volume

Lesion size was visualized using T2-weighted (T2w) magnetic resonance imaging (MRI) on a 3.0T scanner (MAGNETOM Trio; Siemens, Erlangen Germany) in rats undergoing a 7-day STN-HFS or sham stimulation as reported elsewhere [60]. T2w scans were acquired with turbo spin-echo sequences (echo time, 105 ms, repetition time, 2100 ms) and infarct volume was determined using ImageJ Analysis Software 1.45s (National Institutes of Health, Bethesda, MD, USA. http://rsb.info.nih.gov/ij/ (accessed on 29 October 2011)); the hyperintense lesion on each scan (1 mm thick) was traced manually and the areas were then summed and multiplied by the slice thickness.

### 4.11. Immunohistochemistry

After the last experiment, animals were euthanized. Brains were quickly removed and immediately frozen. Coronal sections (10 µm thick) were cut using a sliding microtome (Leica SM 2000R). Sections encompassing the STN were stained with hematoxylin and eosin (H&E) to visualize the anatomic locations of the electrode tip.

### 4.12. Statistical Analyses

Success rates of single-pellet reaching between stimulated and sham-stimulated animals were calculated using the Mann–Whitney U test. Infarct volumes were compared using the unpaired two-tailed Student’s t-test. These two analyses were performed using Prism version 8.0.2 (GraphPad Software, La Jolla, CA, USA). All results are given as mean ± standard error of the mean. A level of *p* < 0.05 was considered significant.

To compare the [^18^F]FDG uptake under two conditions, the respective ratios of the cerebellum-normalized regional PET data were resampled by bootstrapping (10,000 samples) with R (function sample with attribute replace = TRUE) and confidence intervals calculated from the resampled data (*α* = 0.05). Mean differences were expressed as % change and were considered significant if the confidence interval did not include 1.0. Linear mixed-effects modeling and linear regression analysis were performed with the lme4 and base packages in R.

## Figures and Tables

**Figure 1 ijms-23-15862-f001:**
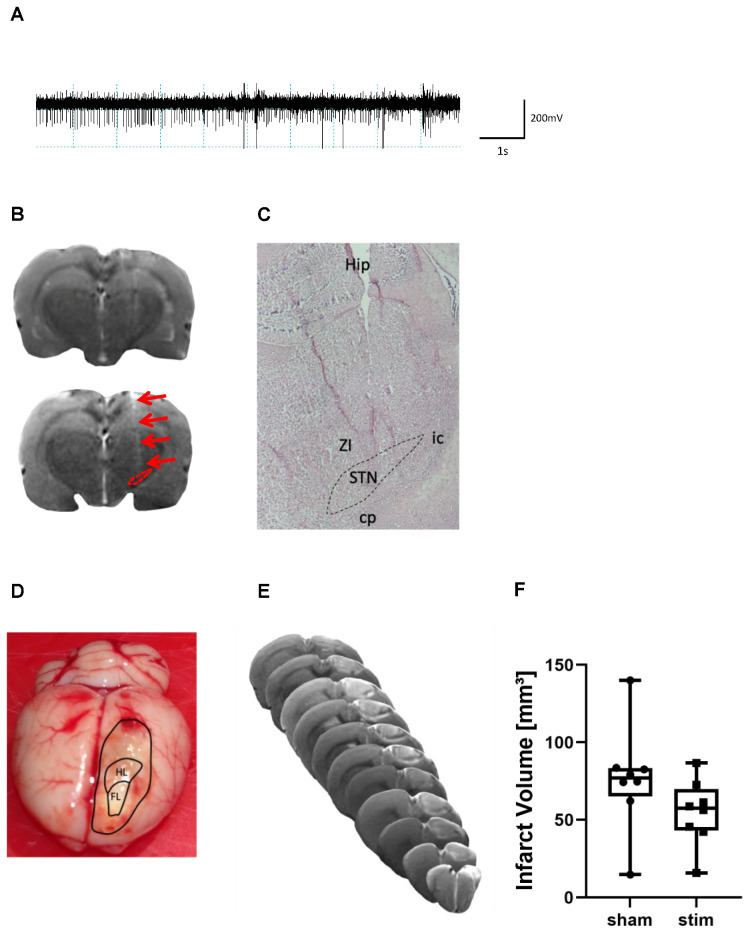
Placement of stimulating microelectrode and determination of infarct size. (**A**) Representative tracing of the typical activity of STN-neurons; (**B**) representative T2w MR scan of a brain slice demonstrating the STN (dashed red circle) as well as the tract of the implanted electrode (red arrows); (**C**) representative HE-stained brain section obtained from the level of the STN demonstrating that the electrode is placed in the STN (magnification: ×40); Zi, zona incerta, Hip, hippocampus, cp, cerebral peduncle, ic, internal capsule; (**D**) macroscopic view of a rat brain after removing from the skull. The outer frame indicates the border of the photothrombotic stroke, and the inner frames correspond with the forelimb (FL) and hindlimb (HL) area according to Paxinos and Watson’s rat brain atlas; (**E**) representative T2w scan of a photothrombotic stroke of a STN-stimulated rat; (**F**) infarct volumes are not significantly different (*n* = 8/group) between the STN-stimulated and sham-stimulated group after stimulation for 7 days. Unpaired, two-tailed Student’s *t*-test.

**Figure 2 ijms-23-15862-f002:**
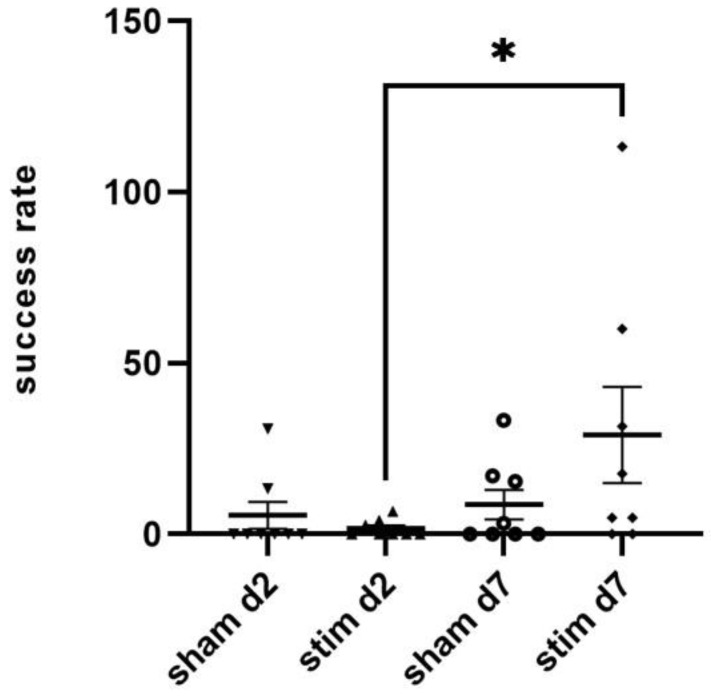
Single-pellet reaching test (SPRT). STN-stimulated animals showed a significant improvement in the SPRT on day 7 when compared with day 2 after intervention, whereas this was not seen in sham-stimulated animals. *n* = 8/group; Mann–Whitney U Test. Stim: stimulated animals; sham: sham-stimulated animals. * *p* < 0.05.

**Figure 3 ijms-23-15862-f003:**
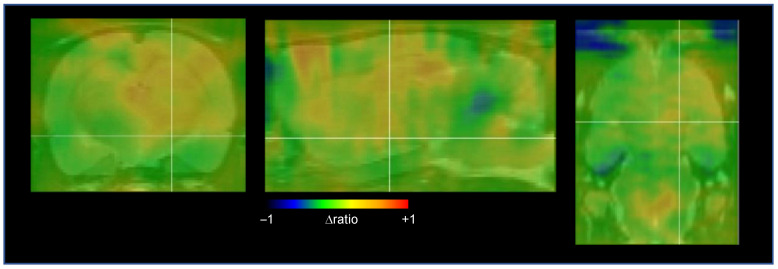
Significant increase in [^18^F]FDG uptake in different brain areas during STN-HFS 3 days after intervention in a coronal (left panel), sagittal (middle panel) and horizontal view (right panel). The panels represent the difference between glucose metabolism with and without STN-HFS. There is an increase of glucose metabolism in the corticosubthalamic and pallidothalamic circuit as well as in the brainstem. PET images are superimposed on an MRI template in gray (MRI template from PMOD software). Δratio, ratio between stimulated and non-stimulated animals, after normalization to the cerebellum.

**Figure 4 ijms-23-15862-f004:**
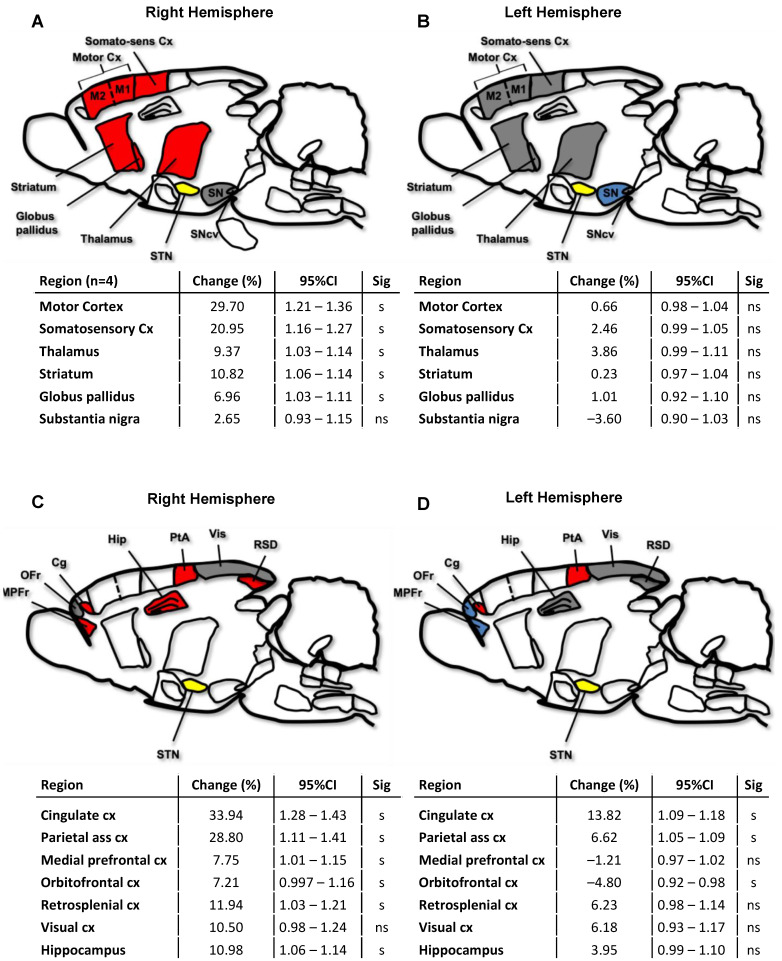
Changes in cerebral glucose metabolism ([^18^F]FDG uptake) after STN-HFS compared to sham stimulation in rats with photothrombotic stroke. In the right hemisphere (**A**), STN-HFS results in a [^18^F]FDG uptake within the corticosubthalamic and pallidosubthalamic loop, whereas no change was seen for these areas in the left hemisphere (**B**). Associative cortices of the right hemisphere (**C**) show a more frequent and stronger [^18^F]FDG uptake compared to the left hemisphere (**D**). When examining locomotor regions, the cerebellar locomotor region (Med) demonstrates a significantly higher glucose metabolism after STN-HFS, whereas the ventromedial hypothalamus (VMH) shows a significant decrease of [^18^F]FDG uptake in both the right (**E**) and left hemisphere (**F**). When applying STN HFS, no change of glucose metabolism was observed in the lateral hypothalamus (LH). STN-HFS results in a significantly higher [^18^F]FDG uptake in most of the examined nuclei of the right-sided brainstem (**G**) and in the Ve, Gi and MdV of the left-sided brainstem (**H**) (*n* = 4). STN, subthalamic nucleus; Cx, cortex; Cn, cuneiform nucleus; PTg, pedunculopontine tegmental nucleus; SN, substantia nigra; SNcv, substantia nigra, compact part, ventral tier; Cg, cingulate cortex; Hip, hippocampus; PtA, parietal association cortex; MPFr, medial prefrontal cortex; OFr, orbitofrontal cortex; RSD, retrosplenial cortex; Vis, visual cortex; VMH, ventromedial hypothalamus; Med, medial (fastigial) nucleus; MdV, ventral part of the medullary reticular nucleus; Sp5, spinal trigeminal nucleus; PCRt, parvicellular reticular nucleus; SpVe, spinal vestibular nucleus; Ve, vestibular nucleus; Gi, gigantocellular nucleus; PnO, pontine reticular nucleus oral part; PMnR, paramedian raphe nucleus; MnR, median raphe nucleus. 95%CI, 95% confidence interval; Sig, significance; s, significant; ns, not significant. Red and blue colored regions demonstrate a significant increase and decrease, respectively, of [^18^F]FDG uptake after STN-HFS; grey-colored regions indicate no change of glucose metabolism during STN-HFS. The STN is colored in yellow.

**Figure 5 ijms-23-15862-f005:**
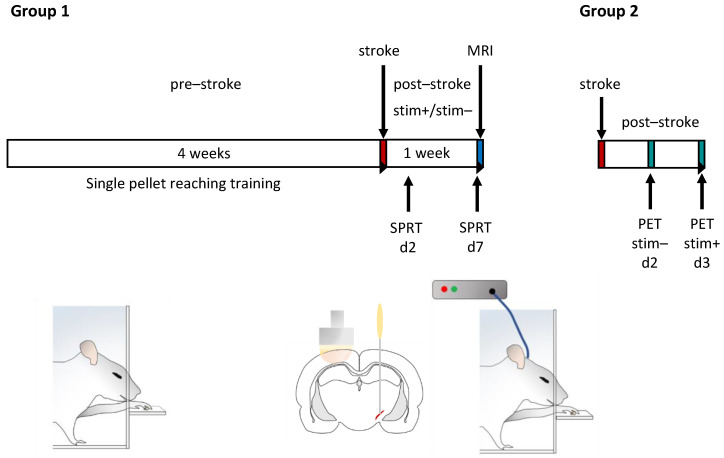
Schematic of the experimental design. Animals were divided into two groups: group 1 was trained in the single-pellet reaching (SPR) test daily (10 min) for 4 weeks. Then, a photothrombotic stroke was induced within the motor forelimb area of the dominant hemisphere and a stimulating electrode was implanted in the ipsilateral subthalamic nucleus (STN). After intervention, a continuous high frequency stimulation (HFS) (*n* = 8) or sham stimulation (*n* = 8) of the STN was started on day 1 after intervention and performed 24 h a day for 7 days. SPR was scored 1 day before as well as on day 2 and day 7 after intervention. Finally, infarct size was assessed using MR imaging on day 7 after intervention. Group 2 (*n* = 6) also underwent photothrombosis and implantation of a stimulating electrode. On day 2 [^18^F]FDG scans of 4 rats were available for analysis. SPRT, single-pellet reaching test; stim−, sham-stimulation stim+, STN-HFS.

## Data Availability

All data generated and analyzed during this study are included in this published article.

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
