# Peer review of "Deep Brain Stimulation in the Subthalamic Nucleus Can Improve Skilled Forelimb Movements and Retune Dynamics of Striatal Networks in a Rat Stroke Model"

_ijms, 2022, doi:10.3390/ijms232415862_

Round 1
Reviewer 1 Report
The manuscript "Deep stimulation of the subthalamic nucleus of the brain can improve skilled forelimb movements and reconfigure striatal network dynamics in a rat model of stroke" focuses on the role of the subthalamic nucleus in the regression of limb paresis after stroke and in the restoration of movement and skills of the forelimbs. The results obtained by the authors open up the possibility of overcoming the neurological deficit in the postischemic period using the method of deep brain stimulation.
But there is a limitation and a drawback in the work.
Unfortunately, stimulated and unstimulated rats are compared on different days after stroke. The difference in cerebral blood flow on the 3rd day after a stroke relative to the 2nd is the result of regression of cerebral edema, and not just stimulation. It is known that cerebral edema significantly decreases by the 3rd day after a stroke.
Abstract part.
19: Authors need to add the area of ​​the brain in which the photothrombic stroke was modeled
23: probably misspelled "STN-HFS" instead "MLR-HFS"
Section Introduction
In my opinion, the introduction lacks an overview of the role of stimulation of the subthalamic nucleus in the treatment of many neurological disorders. Such as Parkinson's disease, essential tremor, dystonia, etc.
Section methods.
87: should indicate when the stimulation was started (hours or minutes) on the first day after the stroke.
154: Add the number of right-handed and left-handed rats.
288: misprint. Repeat twice the values ​​"76.4 288
± 34.2 mm3 54.9 ± 21.2 mm3 vs. 54.9 ± 21.2 mm3"
289: omitted the word "groups" in the phrase ".. in both rats devoted to .."
299: This phrase should be added to the methods section. “Among the stimulated group, 3 rats preferred the right and 4 left paws, while in the sham stimulation group, 4 animals preferred the left paw and 3 preferred the right.” Carefully count the number of rats. The sum is 7 in each group whereas 8 in the methods section.
324, 325: In the methods section the authors should indicate the exact final number of rats that were in the second group (n=4). Details about the death of rats and technical problems only complicate the perception of the results. It is better to remove them.
335: in fig 4. Note should indicate the days after the stroke that were compared.
STN-DBS and STN-HFS are used interchangeably in the text. It is better to choose one type of terminology.
Author Response
Reviewer #1:
- Unfortunately, stimulated and unstimulated rats are compared on different days after stroke. The difference in cerebral blood flow on the 3rd day after a stroke relative to the 2nd is the result of regression of cerebral edema, and not just stimulation. It is known that cerebral edema significantly decreases by the 3rd day after a stroke.
We agree with the reviewer that brain tissue after stroke change over time. However, a photochemically induced brain lesion is characterized by a rapid occlusion of small arterial vessels and by a very small edema compared to the transient or permanent occlusion of the middle cerebral artery (so-called MCAo model). Therefore, we believe that brain tissue on day 2 and 3 after photothrombotic stroke differs only slightly regarding cerebral edema and perfusion.
- Abstract part.
19: Authors need to add the area of the brain in which the photothrombic stroke was modeled.
We thank the reviewer for this helpful comment. We have now indicated the localization of the photothrombotic stroke in the abstract.
23: probably misspelled "STN-HFS" instead "MLR-HFS"
We apologize for this mistake and changed it accordingly.
- Section Introduction
In my opinion, the introduction lacks an overview of the role of stimulation of the subthalamic nucleus in the treatment of many neurological disorders. Such as Parkinson's disease, essential tremor, dystonia, etc.
As suggested by the reviewer, we now added neurological diseases, that may be improved by deep brain stimulation in the subthalamic nucleus.
- Section methods.
87: should indicate when the stimulation was started (hours or minutes) on the first day after the stroke.
Triggered by the reviewer’s comment, we now added the exact time when deep brain stimulation (or sham-stimulation) of the STN has been started (i.e., 24 hours after intervention).
154: Add the number of right-handed and left-handed rats.
Done (please refer also to the comment made to line 299 below).
288: misprint. Repeat twice the values "76.4± 34.2 mm3 54.9 ± 21.2 mm3 vs. 54.9 ± 21.2 mm3"
We apologize for this mistake, the repeated value is now omitted.
289: omitted the word "groups" in the phrase ".. in both rats devoted to .."
Done.
299: This phrase should be added to the methods section. “Among the stimulated group, 3 rats preferred the right and 4 left paws, while in the sham stimulation group, 4 animals preferred the left paw and 3 preferred the right.” Carefully count the number of rats. The sum is 7 in each group whereas 8 in the methods section.
As suggested by the reviewer, we put the aforementioned phrase in the methods section. The number of right- and left-handed rats are now corrected, we apologized for this mistake.
324, 325: In the methods section the authors should indicate the exact final number of rats that were in the second group (n=4). Details about the death of rats and technical problems only complicate the perception of the results. It is better to remove them.
We thank the reviewer for this helpful comment, we now changed this point accordingly.
335: in fig 4. Note should indicate the days after the stroke that were compared.
We thank the reviewer for this important hint, the time of stimulation after induction of stroke is now indicated in the figure legend.
STN-DBS and STN-HFS are used interchangeably in the text. It is better to choose one type of terminology.
As suggested by the reviewer, we have chosen one type of terminology, i.e., STN-HFS.
Reviewer 2 Report
The manuscript is impressive, and the approaches used to test the hypothesis is commendable.
I would like to point out some comments based on functional recovery of stroke rats
Comment1; Why author has chosen one functional recovery test Single pellet reaching tests as there are battery of tests available for functional assessment in stroke.
Functional assessments tests in the rodent stroke models are
Composite Scores |
Assesses a variety of motor, sensory, reflex and balance responses |
Cylinder Test |
Assesses spontaneous forelimb use |
Grid Walking |
Assesses sensorimotor function, motor coordination and placing deficits during locomotion |
Ledged Tapered Beam |
Assesses hindlimb functioning |
Reaching Chamber/Pellet Retrieval |
Assesses skilled forepaw use and motor functioning |
Staircase Test |
Assesses forelimb extension, grasping skills, side bias and independent use of forelimbs |
Pasta Test |
Assesses manual dexterity and fine motor skills |
Ladder Rung Walking test |
Assesses fore- and hindlimb stepping, Placing and coordination |
Forelimb Flexion |
Assesses forelimb function |
Forelimb Placing |
Assesses forelimb function and placing deficits |
Corner Test |
Assesses sensorimotor and postural asymmetries |
Accelerated Rotarod |
Assesses motor coordination and balance |
Adhesive Removal |
Assesses tactile responses and asymmetries |
Morris Water Maze |
Assesses spatial learning and memory |
Radial Arm Maze |
Assesses spatial learning and memory |
These tests have specific function associated to the test. I would like to know that by choosing one behavioral test is able to demonstrate the functional recovery after stroke and will it be able to justify the hypothesis.
Author Response
Reviewer #2:
Why author has chosen one functional recovery test Single pellet reaching tests as there are battery of tests available for functional assessment in stroke. These tests have specific function associated to the test. I would like to know that by choosing one behavioral test is able to demonstrate the functional recovery after stroke and will it be able to justify the hypothesis.
We thank the reviewer for this important comment and the statements regarding the different behavioral tests. After ischemic stroke, around 75% of all stroke survivors suffer from upper limb paresis. In the present study, we aimed to investigate whether STN-HFS improves upper limb motor function, which has been indicated in the manuscript. This was the reason to chose the SPRT, which is a well-established task for examining fine motor control in which rats have to reach, grasp and retrieve a food pellet with the forepaw. We agree with the reviewer that additional behavioral testing would further underpin the predefined hypothesis regarding an improvement of upper limb motor function. In this context, a single behavior test might be a limitation of the study which is now mentioned in the discussion section.